# MAPK Is a Mutual Pathway Targeted by Anxiety-Related miRNAs, and E2F5 Is a Putative Target for Anxiolytic miRNAs

**DOI:** 10.3390/biom13030544

**Published:** 2023-03-16

**Authors:** Javad Amini, Cordian Beyer, Adib Zendedel, Nima Sanadgol

**Affiliations:** 1Department of Physiology and Pharmacology, School of Medicine, North Khorasan University of Medical Sciences, Bojnurd 94149-75516, Iran; 2Institute of Neuroanatomy, RWTH University Hospital Aachen, 52074 Aachen, Germany; 3Department of Biomedicine, University of Basel, 4031 Basel, Switzerland

**Keywords:** anxiety, miRNAs, epigenetic, depression, non-coding RNAs

## Abstract

Anxiety-related disorders (ARDs) are chronic neuropsychological diseases and the sixth leading cause of disability in the world. As dysregulation of microRNAs (miRs) are observed in the pathological course of neuropsychiatric disorders, the present study aimed to introduce miRs that underlie anxiety processing in the brain. First, we collected the experimentally confirmed anxiety-related miRNAs (ARmiRs), predicted their target transcripts, and introduced critical cellular pathways with key commune hub genes. As a result, we have found nine anxiolytic and ten anxiogenic ARmiRs. The anxiolytic miRs frequently target the mRNA of Acyl-CoA synthetase long-chain family member 4 (Acsl4), AFF4-AF4/FMR2 family member 4 (Aff4), and Krüppel like transcription factor 4 (Klf4) genes, where miR-34b-5p and miR-34c-5p interact with all of them. Moreover, the anxiogenic miRs frequently target the mRNA of nine genes; among them, only two miR (miR-142-5p and miR-218-5p) have no interaction with the mRNA of trinucleotide repeat-containing adaptor 6B (Tnrc6b), and miR-124-3p interacts with all of them where MAPK is the main signaling pathway affected by both anxiolytic and anxiogenic miR. In addition, the anxiolytic miR commonly target E2F transcription factor 5 (E2F5) in the TGF-β signaling pathway, and the anxiogenic miR commonly target Ataxin 1 (Atxn1), WASP-like actin nucleation promoting factor (Wasl), and Solute Carrier Family 17 Member 6 (Slc17a6) genes in the notch signaling, adherence junction, and synaptic vesicle cycle pathways, respectively. Taken together, we conclude that the most important anxiolytic (miR-34c, Let-7d, and miR-17) and anxiogenic (miR-19b, miR-92a, and 218) miR, as hub epigenetic modulators, potentially influence the pathophysiology of anxiety, primarily via interaction with the MAPK signaling pathway. Moreover, the role of E2F5 as a novel putative target for anxiolytic miRNAs in ARDs disorders deserves further exploration.

## 1. Introduction

Anxiety is a state of increased attentiveness and receptiveness that results in a range of assessable defensive behaviors. This behavior aims to reduce the damage to the organism in the face of unexpected conditions. This behavior also plays an essential role in all aspects of neural function, shaping and regulating the flow of information through the neural network to produce a specific neural code [1]. However, anxiety dysregulation could be caused by genetic or non-genetic factors such as chronic stress or traumatic brain injury, and these factors lead to anxiety-related disorders (ARDs). ARDs are chronic and disabling diseases, and in the United States alone, at least one in four adult people experiences anxiety in their lifetime [2]. The present diagnostic methods are based on patient self-report measurements and clinical observation. Moreover, the newly proposed Research Domain Criteria (RDoC) system describes anxiety as a phenomenon of multilevel neurobiological features with a range of performance from normal to abnormal behavior. In this regard, a better characterization of anxiety as a complex psychiatric disorder frequently comorbid with depression and alcohol/drug dependence is necessary [3]. In addition, anxiety disorders increase the risk of cardiovascular and high blood pressure diseases associated with premature mortality [4]. Several studies have reported that abnormalities in gene expression and epigenetic factors such as non-coding RNAs, DNA methylation, and histone modification have contributed to the pathogenesis of ARDs [3]. Recently, MS-275 (SNDX-275, a class I and III histone deacetylase inhibitor), Trichostatin A (TSA, a class I and II histone deacetylase inhibitor), and Suberoylanilide hydroxamic acid (SAHA, a pan-histone deacetylase inhibitor) epigenetic drugs are under evaluation in the preclinical phases to help anxious patients [5]. Micro RNAs (miRs) are a very conserved group of small non-coding RNAs that contain 19–25 nucleotides and, in mammals, control 30% of all protein-coding gene activity [6]. Recently, miR have been known as a new regulator of gene expression in the pathophysiology of mental illness. It has been demonstrated that miR contribute to psychiatric disorders such as bipolar disorder, schizophrenia, anxiety disorders, and major depressive disorder [7]. The ability of miRs to simultaneously control the fate of mRNAs and their translation (which frequently act as complimentary sequences to degrade mRNA) predetermines their ability to regulate the activity of whole cellular pathways. It has apparent implications for regulating multifaceted procedures such as neurobehavioral activities [8]. Numerous miR seem to be dysregulated during the pathologic course, and there is substantial evidence supporting the association of miR and their mRNA targets with the severity of neuropsychiatric disorders. It has been reported that miR-483-5p and miR-699p-5p influence anxiety induced by physical or psychological stress via targeting corticotropin-releasing hormone receptor 1 (Crhr1) and Adrenoceptor Alpha 1B (Adra1b) mRNAs [9]. So, identifying crucial upstream hub genes and the epigenetic mechanisms specific for individual anxiety-related miRNAs (ARmiRs) would advance our understanding of how the preconditioning injury induces orchestrated upregulation of anxiety-related genes. This study aims to improve the translatability of studies from animals to humans by introducing the important and experimentally confirmed ARmiR across behavioral dimensions, predicting their function, and evaluating their critical molecular pathways.

## 2. Material and Methods

### 2.1. Searching Methodology

In this study, we used the MeSH terms “Anxiety miRNA”, “Anxiety small RNA”, “Anxiety non-coding RNA”, and “Stress miRNA” as the keywords in the search engines (PubMed, Scopus, and Web of Science) using the same query from November 2012 to November 2022 (last ten years) to obtain ARmiR. We merged the remaining titles across search databases according to the preferred reporting items for systematic reviews and meta-analyses (PRISMA) instruction. The following criteria were considered in this review: original articles written in English considering miRs related to anxiety (confirmed by standard anxiety-like behaviors analysis) with an evaluation of brain samples (in-vivo studies). On the other hand, reviews or any classification other than the original article and articles that did not include anxiety-like behaviors analysis or did not evaluate brain tissue were excluded. After removing duplication and scanning articles, we only selected the miR with a recognized role in anxiety (after reading all abstract articles). Finally, those articles that did not meet the inclusion criteria were eliminated after reading all complete manuscripts. In the reading process, the following information was extracted: change in expression or dysregulation of the miR observed, predicted or confirmed target gene, type of sample and sample size, race and gender of the animal, and behavioral tests for evaluating anxiety. Because the number of studies presenting the exact mechanism of action of each ARmiR was small (low statistical power), we could not perform a meta-analysis.

### 2.2. Target Prediction of Anxiolytic and Anxiogenic ARmiRs

For the target prediction of each ARmiRs reported in the articles included in this review, the miRDB database (http://www.mirdb.org, accessed on 20 March 2022) was used. This database validates ARmiR binding sites on mammalian mRNAs and predicts biologically significant interactions between ARmiR and their mRNA targets [10]. In articles, miR are mentioned without specifying whether 3p or 5p; the variation with more reeds between 3p and 5p was considered. Moreover, the anxiety-related genes (C0003467) were found in the DisGeNET database, which can be accessed at http://www.disgenet.org, accessed on 10 April 2022. DisGeNET is a broad platform that integrates genes and variants associated with human disease [11]. Finally, common target genes for ARmiR were confirmed and compared with anxiety-related genes in each group.

### 2.3. Anxiolytic and Anxiogenic miRs Pathway Analysis and Selection of Influential miRs

The functional interpretation of ARmiR and their targets were evaluated using a database for annotation, visualization, and integrated discovery using Enrichr (https://maayanlab.cloud/Enrichr, accessed on 28 June 2022) [12]. The adjusted *p*-value 0.05 cutoff score was used to determine ARmiRs targets. Lastly, miRs with higher effects in different cellular pathways were highlighted as the most influential ARmiR in each group.

### 2.4. Common Target Genes among Highlighted ARmiRs and Their Interactions

Common target genes among either anxiolytic or anxiogenic highlighted miR are determined via the miRDB database, and their potential protein-protein interaction (PPI) network and key hub genes were constructed using the STRING database version 11.0 (http://string-db.org, accessed on 18 July 2022) with confidence scores of ≥0.9 as a threshold [13,14]. Moreover, specific interactions of each highlighted miR with the mRNA sequence of common target genes were also predicted by the TargetScan database to confirm better the possibility of miR-mRNA interactions (https://www.targetscan.org/vert_80, accessed on 21 August 2022) [15,16]. Besides, the interaction of common target proteins of higher rank with other cellular proteins was predicted by the Pathway Commons database (https://www.pathwaycommons.org, accessed on 5 September 2022) [17].

### 2.5. Determination of Mutual Target Genes and Their Signaling Pathways

Mutual target genes interacting with miRNAs were determined according to the step-by-step analysis, their signaling pathways were introduced, and their relationship with previously confirmed signaling pathways significantly influenced by ARmiRs described based on the information reported in databases. Hub genes were determined, and their related signaling pathways were predicted via the KEGG database in Enrichr (with a high degree and significant *p*-value). Finally, we evaluated the expression pattern of a putative target of anxiolytic/anxiogenic miRs in human tissue via the Human Protein Atlas database (https://www.proteinatlas.org, accessed on 14 November 2022) [18].

## 3. Result

The search combination of all the keywords and including/excluding criteria resulted in a total of n = 744 articles [PubMed (n = 363); Scopus (n = 283 articles), and Web of Science (n = 98 articles)]; 363 manuscripts were removed for duplication, leaving 381 manuscripts. These articles went through qualitative analysis. First, exclusion criteria were applied to these articles, and 185 manuscripts [with the unfitting title (n = 49); reviews and/or case reports (n = 80); in vitro, ex vivo, and in silico studies (n = 56)] were excluded, leaving 196 manuscripts. After the second refined selection, 120 manuscripts were excluded for following these criteria: focusing on miR as a diagnostic biomarker (or risk factor) and/or without proper negative/positive groups (n = 101), with bias in model induction, and/or sample size (n = 19), and 76 manuscripts were left. Finally, after the third selection step, 60 manuscripts were excluded for following these criteria: without appropriate behavioral analysis or study design (n = 32); confounding factors (e.g., the results are the product of another factor, n = 6); and without any effect on anxiety-related behavior (negative consequences, n = 11). After the comprehensive screening, 16 studies were selected, and 21 miRs (nine anxiolytic, ten anxiogenic, and two with dispute function) were obtained from this systematic review (Figure 1 and Table 1).

### 3.1. The Experimentally Confirmed Anxiolytic miRs (Reduced Anxiety)

#### 3.1.1. Let-7d

The human lethal-7 (let-7) family of miR are among the first discovered miRs that are mainly expressed in the different brain parts and are involved in modulating learning and memory. It has been demonstrated that lentiviral-mediated miR let-7d overexpression in the adult mice hippocampus exhibits an anxiolytic and anxiety-like behavior profile (Table 1). The beneficial effect of let-7d against anxiety seems mediated by targeting the dopamine D3 receptor (D3R). Therefore, it was proposed that a behavioral phenotype was also linked to decreased D3R mRNA expression. D3R overexpression, on the other hand, increases anxiety and depression-like behavior as measured by behavioral tests [19].

#### 3.1.2. miR-17/92 Cluster

The miR-17/92 cluster (also known as oncomiR-1) is involved in proliferation, cell cycle, and apoptosis, which are important in neurodevelopment, and is among the first miR to be implicated in a human syndrome (Feingold syndrome, in which individuals have characteristic abnormalities of their fingers and toes). It has been shown that changes in miR-17-92 levels in hippocampal neural progenitors significantly affect neurogenesis and anxiety and depression-related behaviors in adult mice (Table 1). Mice lacking miR-17-92 exhibit anxiety-like behaviors, whereas mice overexpressing miR-17-92 exhibit anxiolytic and anti-depression-like behaviors. The miR-17-92 expression in the adult mouse hippocampus responds to chronic stress, and miR-17-92 rescues hippocampal neural progenitor proliferation defects caused by corticosterone [20].

#### 3.1.3. miR-26a

The miR-26a belongs to the miR26 family (consisting of three subtypes), is highly expressed in dendrites, and is involved in neurite outgrowth and axonal regeneration. It has been reported that miR-26a targets the serotonin autoreceptor, 5-Hydroxytryptamin (5-HT) receptor 1A (HTR1A), which maintains stress resiliency and antidepressant efficacy (Table 1). With antidepressant treatment, miR-26a-2 levels significantly increase in the mouse dorsal raphe nucleus. The transgenic mouse model overexpressed miR-26a-2 in serotonergic neurons and demonstrated increased behavioral resilience to social defeat. On the other hand, the transgenic murine model with a miR-26a-2 knockdown in serotonergic neurons showed increased anxious behavior and a weakened antidepressant response [21].

#### 3.1.4. miR-34b and miR-34c

The miR-34 family members (miR-34a/b/c) are brain-enriched miRs, which are significantly upregulated in the cerebrospinal fluid and brain and target genes possibly involved in cognitive function. NBI27914 hydrochloride is a non-peptide-specific antagonist of the corticotropin-releasing hormone receptor 1 (CRHR1) receptor (Ki = 1.7 nM), which plays a role in trauma-induced anxiety (TIA). The overexpression of miR-34b by a miRNA agomir using a drug delivery system in the paraventricular nucleus reduces hypothalamic-pituitary-adrenal (HPA) axis hyperactivity and anxiety-like behavior (Table 1). TIA decreased by decreasing the hyperactivity of the HPA axis via miR34b targeting CRHR1 [22]. Moreover, overexpression of miR-34c, specifically within the adult central amygdala, induces anxiolytic behavior after challenge (Table 1). One of the miR-34c targets of particular interest is the stress-related corticotropin-releasing factor receptor type 1 (CRFR1) mRNA, which is regulated by a single evolutionary conserved seed complementary site on its 3’ UTR. In vitro studies revealed that miR-34c reduces cell responsiveness to corticotropin-releasing factor in neuronal cells expressing CRFR1 [23]. In contrast, the loss of function of all three members of the miR-34 family (a, b, and c) resulted in a more ‘resilient’ expression phenotype to acute stress-induced anxiety, and facilitation of fear extinction in mice may be due to augmented CRFR1 receptor expression, which inhibits serotonin prefrontal stress-induced release, thus reducing amygdala GABAergic outflow [24].

#### 3.1.5. miR-135a

The miR-135 family consists of two members, miR-135a and miR-135b, and is extremely conserved among mammals. It was reported that the miR-135 family function as oncogenes or tumor suppressor genes in different tissues and physiological conditions, but the exact function of the two miR-135 family members remain largely unknown. The miR-135a modulates neurotransmitter release by regulating synaptic transmission, and anxiety-like behavior increased when the miR-135a was knocked down in the mouse amygdala (Table 1). Based on behavioral studies, electrophysiological experiments in acute brain slices show that miR-135a knockdown increases amygdala spontaneous excitatory postsynaptic currents. In vitro assays and in vivo miRNA overexpression in the amygdala have revealed that complexin-1 and complexin-2, two key regulators of synaptic vesicle fusion, are direct targets of miR-135a [25].

#### 3.1.6. miR-150

As a hematopoietic cell-specific miRNA, miR-150 plays an important role in normal hematopoiesis and hematological malignancies. The miR-150 is significantly reduced in the hippocampi of mice following either acute or chronic restraint stress (Table 1). A study on adult transgenic mice has shown that miR-150 knockout causes anxiety-like behavior in mice according to an open-field test and an elevated plus-maze test. Furthermore, decreased dendrite lengths, dendrite branching, and dendrite spines have been observed in miR-150 knockout mice compared to wild-type mice, and there is a change in the expression of glutamatergic receptors [26].

#### 3.1.7. miR-455

The miR-455-3p is one of the extremely conserved miRs involved in numerous human diseases. The microRNA-455-3p has been identified as a circulating biomarker of early Alzheimer’s disease, with elevated expression in post-mortem brain tissue from Alzheimer’s patients (Table 1). The miR-455 null mice performed significantly worse on the novel object recognition task, indicating deficits in recognition memory and increased anxiety in the open-field test [27].

### 3.2. The Experimentally Confirmed Anxiogenic miRs (Induced Anxiety)

#### 3.2.1. miR-19b-3p

The miR-19b-3p belongs to the miR-17-92 cluster, which possesses oncogenic properties due to its participation in cell survival, proliferation, differentiation, and cell cycle regulation. Drebrin is an F-actin-binding protein that regulates dendritic spine genesis and morphology to modulate memory formation and maintenance. In chronic restraint stress mice, silencing miR-19b-3p expression in vivo or in vitro with an inhibitor increases Drebrin expression, improves the abnormal dendritic structure, and increases spine density in hippocampal CA1 pyramidal neurons (Table 1). By targeting Drebrin, miR-19b-3p upregulation exacerbates chronic restraint stress-induced abnormal synaptic plasticity and cognitive impairment [28].

#### 3.2.2. miR-92a

The miR-92a belongs to the miR-17-92a cluster, and the miR-92a family is a group of highly conserved miRs, including miR-25, miR-92a-1, miR-92a-2, and miR-363. Anxiety behavior is common in Alzheimer’s patients and can be caused by the accumulation of Tau protein, which suppresses the expression of intracellular vesicular γ-aminobutyric acid (GABA) (Table 1). Accumulation of Tau increases miR-92a, which targets the intracellular vesicular GABA transporter (vGAT), and overexpression of vGAT or blocking miR-92a attenuates anxiety [29].

#### 3.2.3. miR-101a-3p

The miR-101a-3p is expressed in many organs and tissues and plays a regulatory role in many human diseases. The miR-101a-3p was discovered to be an abundant miRNA in the amygdala, with significantly higher levels of expression in low-novelty responding rats versus novelty responding rats (Table 1). Overexpression of miR-101a-3p reduced levels of its target, the histone methyltransferase enhancer of zeste homolog 2 (Ezh2), and histone three at lysine 27 (H3K27me3). Using a siRNA approach, researchers discovered that direct knockdown of Ezh2 increased anxiety-like behavior in high-novelty responding rats, but to a lesser extent than miR-101a-3p overexpression [30].

#### 3.2.4. miR-124a

The miR-124 family (miR-124) is highly conserved in animals and regulates neurogenesis by facilitating neural differentiation. After neonatal isolation, adult miR-124a overexpression in the dentate gyrus of the hippocampus significantly exacerbated repetitive behaviors, social impairments, and anxiety-like behaviors (Table 1). High levels of brain-derived neurotrophic factor (BDNF), a direct target of miR-124a, appear negatively correlated with miR-124a expression. BDNF overexpression in the dentate gyrus also reverses the anxiety and autism-like phenotypes caused by neonatal isolation [31].

#### 3.2.5. miR-133a and miR-218

The miR-133 family (miR-133a, miR-133b), transcribed as bicistronic transcripts with miR-1-2, miR-1-1, or miR-206, and is classified as myomiRs for its role in skeletal and cardiac muscle development. miR-218 is expressed throughout the brain, particularly in the hippocampus and prefrontal cortex (PFC) and is debated as a critical candidate for increased stress susceptibility. Anxious people have significantly higher levels of both miR-133a and miR-218 than those controlled in the brain. A study on neuroblastoma cells has shown that miR-133a and miR-218 target synaptic vesicle glycoprotein 2A (SV2A) at both the mRNA and protein levels (Table 1). SV2A is expressed in various GABAergic inhibitory neurons and involves anxiety. The Elevated Plus Maze reveals that conditional knockout mice with decreased SV2A in the hippocampus have elevated anxiety levels [32].

#### 3.2.6. miR-142-5p

The miR-142-5p is a member of the miR-142 miRNA family, which has known roles in cancer, immune diseases, and embryonic stem cells. Single, prolonged stress increases miR-142-5p levels in the amygdala while decreasing levels of its target neuronal PAS domain protein 4 (Npas4), an activity-regulated transcription factor implicated in stress-related psychopathologies (Table 1). Following single prolonged stress, inhibition of miR-142-5p resulted in decreased anxiety-like behaviors and memory deficits, as well as increased expression of Npas4 and BDNF [33].

#### 3.2.7. miR-155

The miR-155 is a multifunctional miRNA enriched in immune system cells and is indispensable for the immune response. According to research on anxiety models, miR-155 knockout mice exhibit reduced anxiety-like responses, depression-associated behavior, and increased affinity for reward (Table 1). Deletion of miR-155 reduces hippocampus inflammation, with lower expression of mRNA encoding the inflammatory cytokines Interleukin 6 (IL-6) and Tumor Necrosis Factor-Alpha (TNF-α). Several genes associated with inflammation, including TNF-α, CCAAT enhancer binding protein beta (Cebpb), the nuclear factor kappa light chain enhancer of activated B cells (NF-kB) pathway, and neuroplasticity factors, including BDNF and transforming growth factor beta (TGF-β), are predicted targets for miR-155, two major pathways that are dysregulated in major depressive disorders [34].

#### 3.2.8. miR-323-3p

The miR-323-3p is one of the brain’s important miR in the pathology of depression. Overexpression of mouse miR-323-3p in the brain was linked to increased anxiety- and depression-like behaviors, whereas decreasing miR-323-3p levels decreased anxiety- and depression-like behaviors (Table 1). MiR-323a-3p influences the expression of erb-b2 receptor tyrosine kinase 4 (ERBB4). This tyrosine-protein kinase receptor, which is highly expressed in the brain, belongs to the epidermal growth factor receptor subfamily and is known to bind to various neuregulins and related growth factors. This gene’s dysregulated function has been linked to schizophrenia and mood disorders [35].

#### 3.2.9. miR-494

The miR-494-3p is a member of the miR-494 family, and it has been reported that it could act as an oncomiR. The amygdaloid miR-494 plays a critical role in ethanol’s anxiolytic-like effects (Table 1). An in vivo infusion of an antagomir against miR-494 into the amygdala’s central nucleus was able to mimic the anti-anxiety behavioral effects of ethanol and increase the expression of CBP/p300-interacting transactivator 2 (Cited2), CREB binding protein (CBP) and p300. MiR-494 regulates chromatin structure by modulating histone acetylation in the central nucleus of the amygdala and regulates acute ethanol and its behavioral effects [36].

### 3.3. The miRs with Dispute Function (Anxiolytic/Anxiogenic)

The miR-212/132 family had relatively high expression in the brain and regulated neural outgrowth under the effect of nerve growth factors. The miR-132 and miR-212 dysregulation contribute to abnormal neuronal plasticity and gene expression in the mammalian brain (Table 1). Using an acute-stress model, researchers discovered that both miR-132 and miR-212 levels rise in the wild-type murine hippocampus and amygdala. In contrast, significant increases in anxiety-like behaviors were observed in both the miR-132 overexpression and knockout lines. Furthermore, in the hippocampus and amygdala of miR-132/212 conditional knockout and miR-132 transgenic mice, expression of sirtuin 1 (SIRT1) and phosphatase and tensin homolog (Pten), two miR-132 target genes implicated in the regulation of anxiety, was differentially regulated [37].

### 3.4. Construction of Common ARmiRs Gene Targets and Anxiety-Related Genes

Three genes (all involved in the lipid metabolism), including fatty Acyl-CoA synthetase long-chain family member 4 (Acsl4), AF4/FMR2 family member 4 (Aff4), and Kruppel like factor 4 (Klf4) by five interactions, were the common gene target for anxiolytic miRs (Figure 2A). The trinucleotide repeat containing adaptor 6B (Tnrc6b), clock circadian regulator (Clock), putative homeodomain transcription factor 2 (Phtf2), argonaute RISC component 1 (Ago1), F-box protein 30 (Fbxo30), ionotropic glutamate receptor AMPA type subunit 2 (Gria2), solute carrier family 30 member 7 (Slc30a7), sodium voltage-gated channel alpha subunit 2 (Scn2a), and HIVEP zinc finger 2 (Hivep2) are common genes targeted by the anxiogenic miRs (Figure 2B). Tnrc6b and Clock by 8 and 7 interactions are the most common gene targets for anxiogenic miRs (Figure 2B). All common genes (anxiolytic/anxiogenic) are evaluated as anxiety-related genes by DisgeNet data. Based on DisgeNet data, Acsl4, Clock, Scn2a, and Hivep2 are confirmed as anxiety-related genes (Figure 2A,B).

### 3.5. Construction of Cellular Pathways of ARmiRs Targets

Common cellular pathways targeted by ARmiR can help to understand the primary function of these miR. Three critical cellular pathways that anxiolytic miRs significantly influenced were mitogen-activated protein kinase (MAPK), phosphatidylinositol-3 kinase (PI3K)-protein kinase B (PKB, or Akt), and autophagy signaling pathways (Figure 2C). Many genes targeted by anxiolytic miRs belong to MAPK and PI3K-Akt, and they are targeted by six and seven anxiolytic miRs, respectively (Figure 2C). Moreover, three important cellular pathways that anxiogenic miRs significantly influenced were AMP-activated protein kinase (AMPK), MAPK, and axon guidance signaling pathways (Figure 2D). Many genes targeted by anxiogenic miRs belong to AMPK and MAPK, and all and four anxiogenic miRs target them, respectively (Figure 2D).

### 3.6. Cellular Pathways and Common Targets of Anxiogenic miRs

Therefore, three important anxiogenic miRs with higher effects in the cellular pathways (miR-19b, miR-92a, and miR-218) were selected, and their common mRNA targets were predicted via the miRDB database (Figure 3A). We found 14 common targets for this highlighted anxiogenic miRs, among them Solute carrier family 17 (vesicular glutamate transporter), member 6 (Slc17a6), Wiskott-Aldrich syndrome-like (Wasl), and Ataxin-1 (Atxn1) have a higher rank for interaction (Figure 3B). Remarkably, none of these proteins interacted with other proteins in this category after PPI evaluation in the STRING database (Figure 3B). Moreover, the pathway commons database also depicts the possible interaction of SLC17a6, WASL, and ATXN1 with cellular proteins (Figure 3C). Finally, results from analysis of signaling pathways via the KEGG database in Enrichr have shown that SLC17a6 is involved in the synaptic vesicle cycle, WASL is engaged in the adherent junction, and ATXN1 is involved in the Notch signaling pathway. Together, according to our new findings and previous evidence, we represented the schematic view of the functional role of anxiogenic miRs via targeting SLC17a6, WASL, and ATXN1 signaling pathways and the possible interactions (Figure 4). Based on our data, we could not introduce any interaction between the SLC17a6, WASL, and ATXN1 signaling pathways and their connection with the MAPK signaling pathway in the cell (Figure 4).

### 3.7. Cellular Pathways and Common Targets of Anxiolytic miRs

Three important anxiolytic miRs with higher effects in the cellular pathways (Let-7d, miR-17, and miR-34c) were selected, and their common mRNA targets were predicted via the miRDB database (Figure 5A). We found eight common targets for this highlighted anxiolytic miRs among them, only E2F transcription factor 5 (E2F5) has a higher rank for interaction (Figure 5B). Remarkably, none of these proteins showed interactions with other proteins in this category after PPI evaluation in the STRING database (Figure 5B). Moreover, the possible interactions of E2F5 with cellular proteins are also depicted via the pathway commons database (Figure 5C). Finally, results from analysis of signaling pathways via the KEGG database in Enrichr have shown that E2F5 is involved in the transforming growth factor beta (TGF-β) signaling pathway. Together, according to our new findings and previous evidence, we re-drew the schematic view of the functional role of anxiolytic miRs by targeting the E2F5 signaling pathway and possible interactions with PI3K-Akt and MAPK signaling cascades (Figure 6). Our data shows that E2F5/c-*Myc* interaction plays a critical role in the connection between TGF-β and MAPK, the signaling pathway, and the regulation of the broader spectrum of downstream effector molecules in the cell (Figure 6).

### 3.8. Distribution of Target Genes Expression Patterns in the Brain and Human Body

Our finding determined that E2F5 acts as a hub target gene for anxiolytic miRs, and its effect may be mediated by its interaction with c-Myc and impact on other cellular pathways like MAPK and further suppression/activation of key regulators in anxiety-related behavior (Figure 6). The exact interaction region of each anxiogenic miRs with mentioned genes mRNA is confirmed with the TargetScan database except for miR-19b/Wasl interaction (Figure 7). On the other side, the exact interaction region of each anxiolytic miRs with mRNA of E2F5 is also confirmed with the TargetScan database (Figure 7). Furthermore, the cerebral cortex is the brain region most critically involved in behavioral flexibility [38], and an increasing number of studies point to the involvement of changes in its function in anxiety disorders [39].

So, using the Human Protein Atlas database, we reported that E2F5 expresses in all parts of the brain; however, it is not a noticeable sharp differentiation (Figure 8A). On the other hand, evaluation of hub target genes for anxiolytic miRs show that Atxn1 and Was1 have higher expression levels in the cerebral cortex and hippocampus, and Slc17a6 mainly expressed in the thalamus, pons, and medulla oblongata regions (Figure 8A). Finally, according to the result of the protein atlas database, E2F5 expression could be observed in several parts of the human body, especially in bone marrow/lymphoid tissues (male) and endocrine tissues (female) of normal individuals (Figure 8B).

## 4. Discussion

Anxiety disorders are the most common type of mental illness worldwide and are estimated to affect 22.2 percent of Americans and 14 percent of Europeans yearly [40]. Micro RNAs (miRs) are a highly preserved group of small non-coding RNAs (sncRNAs) that regulate neurobiological systems. The miR represent an exciting frontier to be explored in the etiology and treatment of anxiety and stress-related disorders due to their abundance in the central nervous system and their involvement in synaptic plasticity and neuronal differentiation [41]. Although some miRs, such as miR-let-7b, miR-17, and miR-21, are expressed ubiquitously, there are several miRs, such as miR-128b, that are enriched in the brain [42], as well as miR that are described in a temporal (miR-7), regional (miR-34a), and cell-type-specific manner (miR-132) in the brain [43,44].

We find that the mRNAs of the Acsl4, Klf4, and Aff4 genes are the typical target genes for the anxiolytic miRs (Figure 2A). The long-chain family of acyl-CoA synthetase proteins, which includes ACSL4, has recently been demonstrated to have a significant role in ferroptosis, and ACSL4 is a critical player in metabolic diseases. Previous research has discovered that metabolic disorders in amino acid synthesis, lipid synthesis, and iron transport cause cell death [45]. ACSL4 upregulates in chronic unpredictable mild stress, while it is downregulated by fluoxetine as an antistress drug [46]. KLF4 protein is a zinc finger-containing transcription factor that regulates various cellular processes, including cell growth, proliferation, and differentiation [47]. KLF4 transactivated the cyclin-dependent kinase inhibitor 1A (CDKN1A) promoter, where CDKN1A is involved in cell apoptosis and upregulated in the hippocampus of anxious mice [48,49]. Through SIRT1/NF-κB/p53 signaling, KLF4 activates inflammatory injury and oxidative stress in LPS-induced ATDC5 (chondrogenic cell line) cells [50]. Studies on inflammation and oxidative stress have shown that KLF4 may cause anxiety through changes in the physiology of the brain [51,52]. AFF1, AFF2, AFF3, and AFF4 (paralogs of the ALF transcription factor) are members of the transcriptional super elongation complex, which controls the expression of genes involved in neurogenesis and development. It has been demonstrated that mutations of the fruit fly ALF orthologous gene Lilli (Lilli) prevent neuronal differentiation and reduce cell size and growth [53].

Moreover, we also find that the mRNAs of nine genes (Tnrc6b, Clock, Gria2, Scn2a, Phtf2, Hivep2, Ago1, Fbxo30, and Slc30a7) are the typical targets for the anxiogenic miRs (Figure 2B). A unique genetic condition caused by TNRC6B variants includes developmental delay, intellectual disability, autism, attention deficit disorder, hyperactivity disorder, other behavioral disorders, and recurring neurocognitive and behavioral symptoms [54]. In mammals, circadian rhythms are produced by a molecular oscillator in which the expression of the genes encoding the CRYPTOCHROME (CRY) and PERIOD (PER) proteins is regulated by the circadian locomotor output cycles protein kaput (CLOCK)-brain and muscle ARNT-like protein 1 (BMAL1) transcription factors [55]. A study on mouse models has shown that inhibition of cyr1/2 (Cry1^−/−^Cry2^+/+^ and Cry1^+/+^Cry2^−/−^) impairs recognition memory and elevates anxiety [56]. GRIA2 encodes for a subunit of the AMPA-sensitive glutamate receptor (GluA2), an essential component of excitatory synaptic transmission and a ligand-gated ion channel in the central nervous system. The promoter of GRIA2 methylates in schizophrenia, and mutations in GRIA2 cause childhood-onset schizophrenia [57]. Neurological conditions like epilepsy, autism spectrum disorders, intellectual disability, and schizophrenia have been linked to mutations of the Scn2a gene, which codes for the voltage-gated sodium channel alpha-II subunit Nav1.2. Moreover, mice with the Scn2a^KO/+^ mutation have various symptoms frequently seen in models of schizophrenia and autism spectrum disorder [58]. In the large zinc-finger transcriptional protein family (ZnF-C2H2 type), implicated in immune responses, adipogenesis, bone morphogenesis, and brain development, HIVEP2 encodes the human immunodeficiency virus type I enhancer binding protein 2 [59]. A number of the genes controlled by HIVEP2 are connected to brain development, and functional studies have suggested that HIVEP2 plays a role in neurodevelopment. In addition, working memory issues, elevated anxiety, and hyperactivity are all present in HIVEP2-knockout mice [60]. AGO proteins control a cascade of signals in post-transcriptional gene silencing (PTGS) through translational repression, deadenylation, and mRNA degradation. Interestingly, Kowalczyk and their colleague demonstrated a lower risk of depression with the polymorphic variant of the rs4961280/AGO2 gene (genotypes C/A and C/A-A/A) in the Polish population [61]. FBXO30 is an E3 ubiquitin ligase and targets RARγ for ubiquitination and proteasomal degradation. Bone morphogenic proteins (BMP) are essential signaling molecules used throughout neural development to regulate the complex procedures necessary to produce a functional central neuron system, which they positively regulate by Fbxo30 in mammalian cells [62].

As we summarized here, numerous ARmiRs target components of the PI3K-Akt, AMPK, and MAPK signaling cascades, suggesting that the dysregulation of these systems might be a common denominator of ARDs. Most interestingly, MAPK is a mutual pathway targeted by anxiety-related miRNAs. MAPK pathways are crucial for every aspect of cell survival and growth, and control fundamental cellular processes (including proliferation, migration, differentiation), embryogenesis, and cell death. So far, 14 MAPKs divided as seven groups (may further be classified into conventional or atypical MAPKs) have been recognized in mammals. Conventional MAPKs include extracellular signal-regulated kinase (ERK)1/2, ERK5, Jun N-terminal kinase (JNK)1/2/3, p38 isoforms α/β/γ(ERK6)/δ while atypical MAPKs comprises of ERK3/4, ERK7, and Nemo like Kinases (NLK) [63]. MAPK pathway is strongly regulated by phosphatases and bidirectional communication with other pathways (such as the Akt/AMPK pathway) under normal conditions, and its signaling nodes can have different functions. For example, a MAPK/ERK signaling node can function as a tumor suppressor and the more common pro-oncogenic signal [64]. The main outcome depends on the signal intensity and the context or tissue in which the signal is aberrantly activated. So, miRNAs could have unique effects in different conditions, targeting specific signaling nodes in this pathway [65]. Moreover, one miRNA can exert other effects (targeting more than a particular molecule). The miRNAs usually result in either translational repression or the degradation of the target mRNA (via binding to specific sequences at the 3′-UTR) or the inhibition of gene expression (via binding to either the 5′-UTR or the coding region of the target molecules). In contrast, their binding at the promoter site of the target molecules enhances gene expression [66]. In summary, a single miRNA could potentially target numerous mRNAs, while at the same time, one mRNA could contain multiple binding sites for miRNAs, turning this into a possibility that a vast number of biological processes could be regulated by this interaction [67]. Therefore, more studies are needed to prove that targeting the same pathway (AMPK) by different ARmiRs (anxiolytic and anxiogenic) could lead to the same or different outcomes.

On the other hand, concerning the ARDs, these signaling cascades remarkably take part in regulating glucose homeostasis. However, no specific treatment for the ARDs based on glucose homeostasis (metabolic syndrome) targeting these protein kinases has yet been developed. Studies on mouse models of anxiety have shown that improving anxiety with Cananga odorata essential oil downregulates ERK1/2 phosphorylation levels in the prefrontal cortex and hippocampus [49], and the MAPK signaling pathway is associated with anxiety. Anxiety and fear are expected during the withdrawal stage and play a significant role in establishing alcohol dependence. The PI3K-AKT-GSK3β signaling pathway is activated after alcohol withdrawal, and the downstream cAMP response element-binding protein (CREB) phosphorylation is increased. However, dampening this situation with medicine reverses PI3K-AKT-GSK3β-CREB pathway phosphorylation [68]. As ARmiRs influence several cellular pathways, we selected pathways with a maximum number of changed genes to understand better the more targeted pathways in each category (anxiolytic and anxiogenic). On the other hand, we selected those ARmiRs that first had more effect on these selected pathways and secondary had a larger range of effects via targeting several different signaling pathways. So, with this screening approach, we reached a short list of ARmiRs that have maximum impact on all signaling pathways and also have top common targeted pathways. Among common targets for highlighted anxiogenic miRs, Wasl, Atxn1, and Slc17a6 showed higher ranks for interaction.

The Wasl can be found in all subsets of myeloid cells in the brain, albeit at different levels of expression, and it is crucial for regulating microglial morphology, phagocytosis, and migration [69]. The Notch signaling pathway includes BOAT1 and ATXN1. Both ATXN1 and BOAT1 bind to the Hey1 promoter region in mammalian cells and, through interactions with the transcription factor CBF1, which is essential for the Notch pathway, suppress the transcriptional output of Notch. Additionally, ATXN1 is necessary for processing -amyloid precursor protein, motor coordination, and cognitive function [70].

Among the common targets for highlighted anxiolytic miRs, only E2f5 showed a higher rank for interaction. Here, we have delineated E2f5/c-Myc interaction as an interactive signaling network linking TGF-β, and MAPK, signaling pathways together during anxiety regulated by anxiolytic miRs. The E2F family of transcription factors (seven members in mammals) regulates cell cycle progression, especially in S-phase entry where E2F1, E2F2, and E2F3 have potent transcriptional activation activity, interact exclusively with pRb [71]. In contrast, E2F4 and E2F5 are poor transcriptional activators and appear to function as repressors by recruiting pocket proteins to E2F-regulated promoters [72]. Furthermore, E2F5 was highly expressed in glioblastoma tumors, and it has been reported that miR-Let-7c and miR-129-3p inhibited glioma development by targeting E2F5. Remarkably, E2F5 was significantly correlated with M1 macrophage markers (IRF5) and monocyte markers (CD115), suggesting that it plays a role in regulating monocyte polarization in glioblastoma [73,74].

It has been suggested that c-Myc plays a critical role in brain regeneration by regulating the expression of a spectrum of downstream regeneration-associated genes (RAGs) and subsequent effector molecules [75]. A proteomics study reported that overexpression of c-MYC promoted axonal regeneration after optic nerve injury and recognized it as an injury signal hub in the retinal ganglion cells [76]. Another study stated that c-MYC regulates sensory axon regeneration via its downstream effectors, telomerase reverse transcriptase, and p53 [77]. However, a potential mechanism by which E2F5 and/or c-MYC regulate anxiety-related disorders remain unknown, and the role of E2F5/c-MYC interaction in regulating downstream genes involved in anxiety has not been addressed so far. It will be interesting for upcoming studies to explore the role of highlighted anxiolytic miRs (Let-7d, miR-17, and miR-34c) in the expression of E2F5/c-MYC in different brain regions (hippocampus, prefrontal cortex, amygdala, striatum) and cell-type (inhibitory neurons, pyramidal neurons, and glia) specific manners using cutting-edge techniques. Shortly, there is much hope that a deeper understanding of the mechanism of action of ARmiRs will yield more significant insights into the specific function of these ARmiRs and underlying molecular mechanisms.

## 5. Conclusions

Here we summarize the evidence supporting the role of miRs in the onset/progression or control of ARDs. Interestingly, three highlighted anxiolytic miRs (Let-7d, miR-17, and miR-34c) were the most promising for treating anxiety because they shared the same hub gene and signaling pathway. Furthermore, we showed that E2F5/c-Myc interaction could be a central linker between TGF-β/MAPK pathways in the glial cells. Our results provided evidence of the correlation between anxiolytic miRs and E2F5/c-MYC signaling that could be useful for disease diagnosis based on future in-depth analyses of large case numbers and cohort studies. Evaluation of up- or downregulation of these candidate ARmiRs (anxiolytic and anxiogenic) in essential nodes of the anxiety neurocircuitry would indeed confirm their effect on anxiety-related behavior in humans. In addition, future studies suggest the evaluation of the connecting role of ARmiRs and metabolic signaling pathways (especially MAPK) by behavior and functional studies in animal models. This will be commanding to progress the use of ARmiRs in diagnosing and treating ARDs.

## Figures and Tables

**Figure 1 biomolecules-13-00544-f001:**
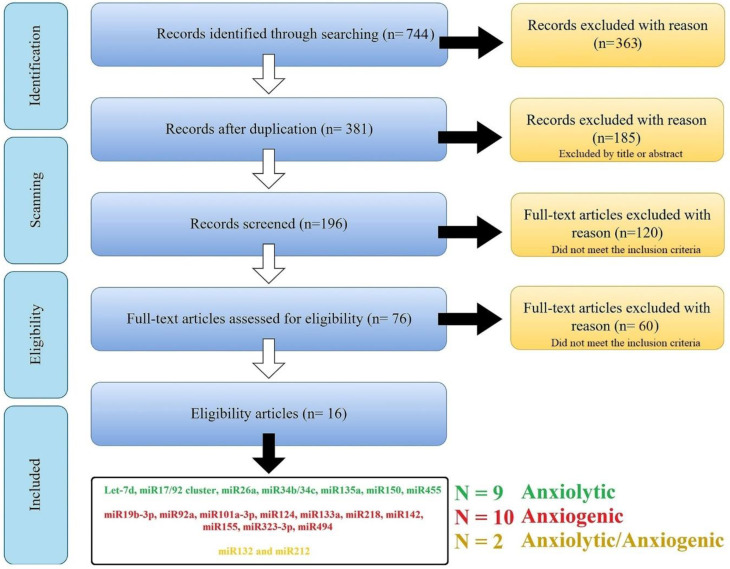
An overview of searching and scanning articles according to the PRISMA. The search of the PubMed, Scopus, and Web of Science databases provided 744 citations. Of these, 363 studies were removed after filtering out the duplicates, and of those, 185 studies were discarded after reviewing the titles. After adjusting for abstracts that did not meet the criteria, 196 articles remained. Finally, the full text of the remaining 196 citations was examined in more detail, and 120 studies in the first step and 60 in the second did not meet the inclusion criteria described. After the comprehensive screening, we reach twenty-one miRs from sixteen articles.

**Figure 2 biomolecules-13-00544-f002:**
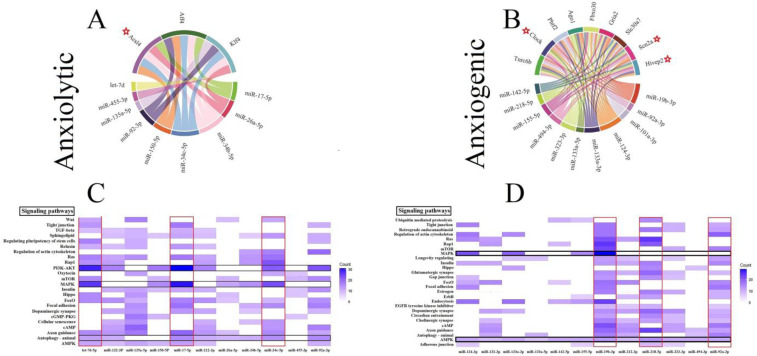
Common gene targets of ARmiRs and their signaling pathways. (**A**) The anxiolytic miRNAs mainly target three genes, including Acs14, Aff4, and K1f4. (**B**) There are nine common gene targets for anxiogenic miRNAs, and the highest interactions were by Tnrc6b (8 interactions). (**C**) Anxiolytic miRNAs cellular pathways. MAPK and PI3K-Akt are the most common target cellular pathways for anxiolytic miRNAs. The color intensity and scale introduce the number of genes each miRNA targets and participates in each cellular pathway. (**D**) Anxiogenic miRNAs cellular pathways. AMPK and endocytosis are the main target cellular pathways for the anxiogenic miRNAs. The color intensity and scale introduce the number of genes each miRNA targets and participates in each cellular pathway. Red Stars indicated genes with maximum interaction with miRNAs.

**Figure 3 biomolecules-13-00544-f003:**
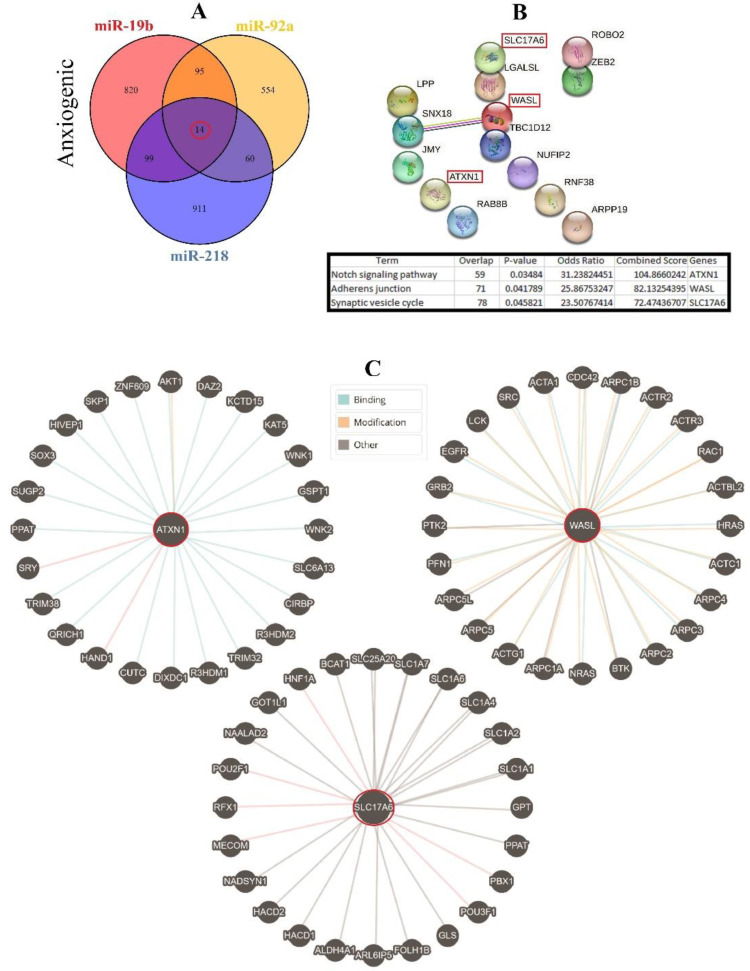
Common gene targets of highlighted anxiogenic miRNAs (miR-19b, miR-92a, and 218), their PPI, and protein targets. (**A**) Target prediction of highlighted anxiogenic miRNAs and selection of shared genes. (**B**) PPI of 14 common genes and selection proteins with a higher rank (SLC17A6, WASL, and ATXN1). Hub genes were determined via the KEGG database in Enrichr, and information is indicated in a table. (**C**) Protein interaction of anxiogenic’s common gene targets with other proteins.

**Figure 4 biomolecules-13-00544-f004:**
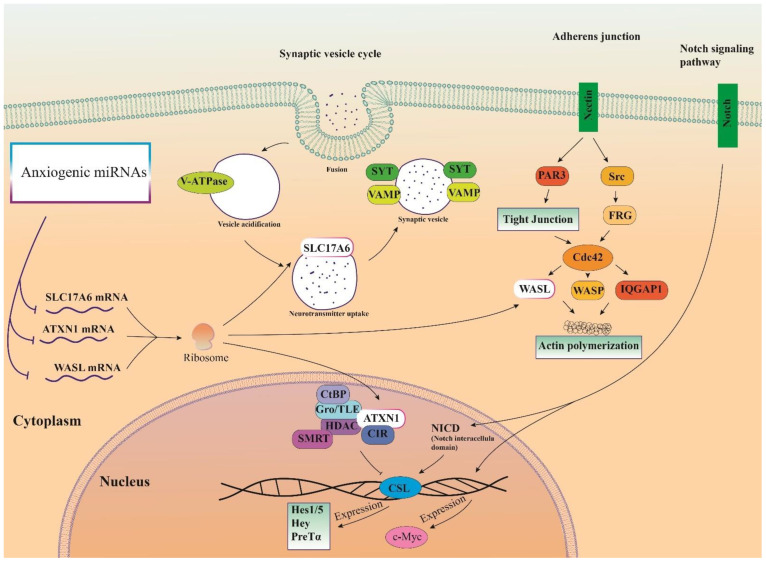
Illustration of the signaling pathways of highlighted anxiogenic miRNAs (miR-19b, miR-92a, and 218). Genes were targeted by highlighted anxiogenic miRNAs participating in Notch signaling, adherence junction, and synaptic vesicle cycle signaling pathways.

**Figure 5 biomolecules-13-00544-f005:**
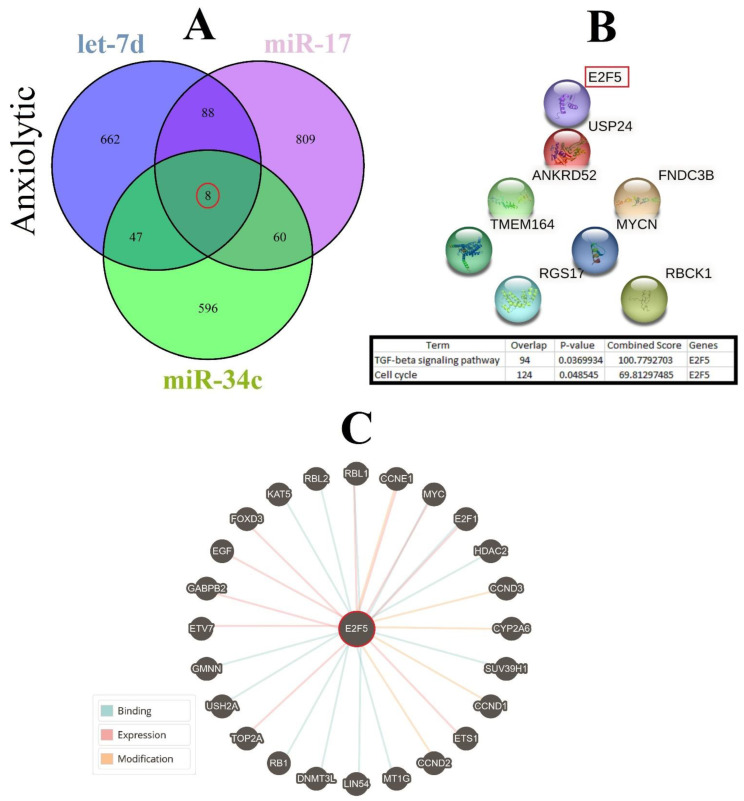
Common gene targets of highlighted anxiolytic miRNAs (miR-34c, Let-7d, and miR-17), their PPI, and protein targets. (**A**) Target prediction of highlighted anxiolytic miRNAs and selection of shared genes. (**B**) PPI of the common eight genes and selection proteins with a higher rank (E2f5). Hub genes were determined via the KEGG database in Enrichr, and information is indicated in a table. (**C**) Protein interaction of anxiolytic’s common gene targets with other proteins.

**Figure 6 biomolecules-13-00544-f006:**
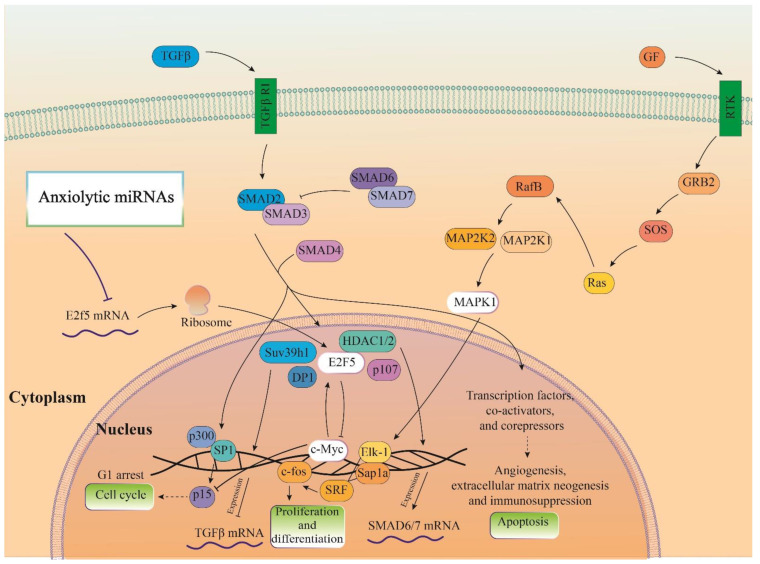
Illustration of the signaling pathways of the highlighted anxiolytic miRNAs (miR-34c, Let-7d, and miR-17). Gene was targeted by highlighted anxiolytic miRNAs (E2f5) that participate in TGF-β signaling and are also connected with MAPK and PI3K-Act signaling pathways via interaction with c-Myc.

**Figure 7 biomolecules-13-00544-f007:**
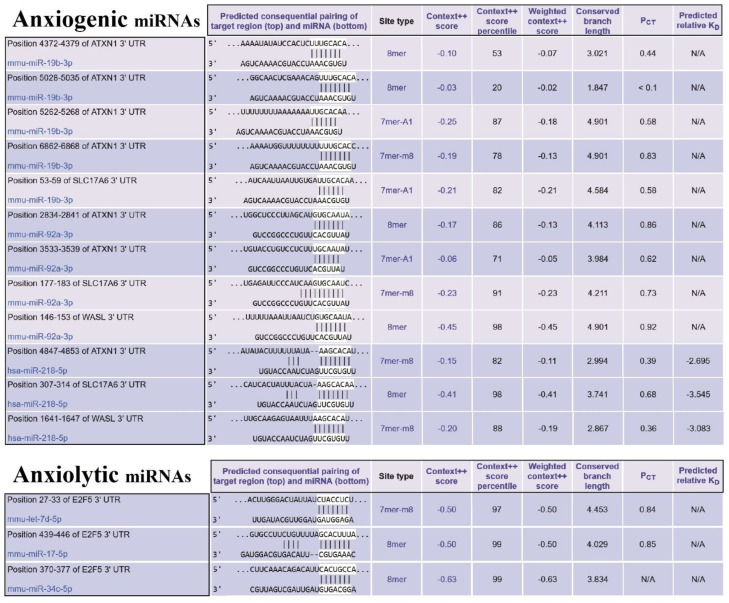
The miRs/mRNAs interactions and related information. (**Upper table**) The sequence of miRNAs of three selected essential proteins (SLC17A6, WASL, and ATXN1) at 3′-UTR and their interaction with anxiogenic miRNAs (miR-19b, miR-92a, and 218). (**Lower table**) The sequence of miRNAs of selected important protein (E2f5) at 3′-UTR and their interaction with anxiolytic miRNAs (miR-34c, Let-7d, and miR-17).

**Figure 8 biomolecules-13-00544-f008:**
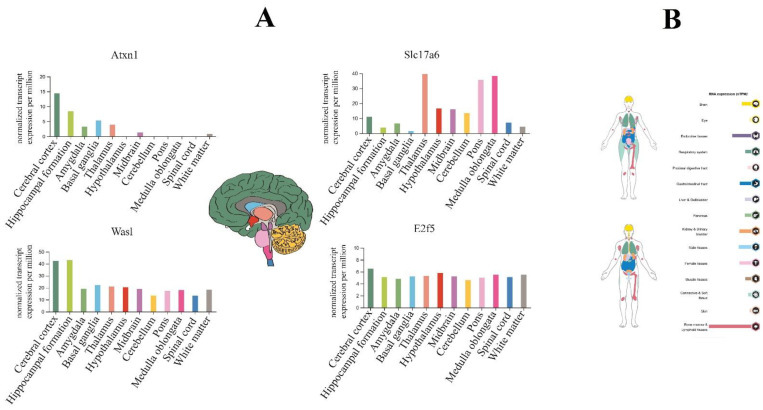
The brain tissue expression of highlighted genes targeted by anxiogenic (miR-19b, miR-92a, and 218) and anxiolytic (miR-34c, Let-7d, and miR-17) miRNA. (**A**) The colored bar plot compares the expression of genes in the brain based on normal transcription per million (nTPM). E2f5 expression in all brain parts, especially in the cerebellar cortex, which includes most of the brain (dark green color area). (**B**) overview of the expression of E2f5 (up image devoted to the female body and below to the male body); it highly expresses in bone marrow and lymphoid tissues, and endocrine tissues.

**Table 1 biomolecules-13-00544-t001:** List of anxiolytic (9), anxiogenic (10), and other anxiety-related miRNAs.

	Name	Analyzed Tissue (Animal)	Proposed Mechanism of Action	Behavioral Test	Ref.
**Anxiolytic**	Let-7d	Hippocampus (C57BL/6, M mice)	Anxiolytic and anti-depressant-like action through targeting of D3R in anxiety	Elevated plus maze, Open field	[19]
	miR-17 miR-92	Amygdala, Prefrontal cortex, Hypothalamus (C57BL/6, M mice)	Exhibit anxiolytic and anti-depression-like behavior by regulating Sgk1	Elevated plus maze, Open field, Restraint stress, forced swim, Tail suspension, Sucrose preference	[20]
	miR-26a	Dorsal raphe nucleus (C57BL/6, M mice)	Functions as an antidepressant by targeting HTR1A in serotonergic neurons	Elevated plus maze, Dark-light transfer, Open field, Forced swim	[21]
	miR-34b	The paraventricular nucleus (M Wistar Rat)	Targeting CRHR1 and attenuating trauma-induced anxiety by decreasing the hyperactivity of the HPA axis	Elevated plus maze, Open field	[22,23]
	miR-34c	Amygdala (C57BL/6, M mice)	Reduces the responsiveness of cells to CRF in CRFR1-expressing neuronal cells	Elevated plus maze, Dark-light transfer, Open field	[24]
	miR-135a	Amygdala (C57BL/6J, M mice)	Targets complexin-1 and 2 and modulate presynaptic glutamate neurotransmission	Elevated-plus maze	[24,25]
	miR-150	Hippocampus (C57BL/6, M mice)	Decrease anxiety-like behavior by influencing the synaptic plasticity	Elevated plus maze, Open field	[26]
	miR-455	Hippocampus (C57BL/6J, M mice)	Decrease anxiety and increase recognition memory in Alzheimer	Open field, Novel object recognition	[27]
**Anxiogenic**	miR-19b-3p	Hippocampus (C57BL/6J, M mice)	Exacerbates CRS-induced cognitive impairment by targeting Drebrin	Elevated plus maze, Open field	[28]
	miR-92a	Hippocampus (C57BL/6J, M mice)	Targeted vGAT mRNA 3′ UTR, inhibited its translation, and increased anxiety in Tauopathy	Elevated plus maze, Open field, social interaction, Unchanged motor function	[29]
	miR-101a-3p	Amygdala (HR/LR, M rats)	Increases anxiety-like behavior at least partially via repression of Ezh2	Elevated plus maze, Open field	[30]
	miR-124a	Dentate gyrus (M/F Wistar rats)	Through its target, BDNF may influence neonatal isolation-induced anxiety-like behaviors	Elevated plus maze, Open field	[31]
	miR-133a miR-218	Amygdala (Postmortem, H)	Targets the 3′UTR of SV2A and increases anxiety-like behavior	Self-reported	[32]
	miR-142-5p	Amygdala (M Sprague-Dawley rats)	Targets Npas4 and increases anxiety-like behavior	Elevated plus maze, Open field, Morris water maze	[33]
	miR-155	Hippocampus (C57BL/6J M/F mice)	Increases anxiety-like behavior and inflammatory cytokines (IL-6 and TNF-α)	Elevated plus maze, Open field, Forced swim	[34]
	miR-323-3p	Anterior cingulate cortex (M CD-1 mice) Anterior cingulate cortex and lateral habenula (Postmortem, H)	Targets ERBB4 and elevates anxiety-like behaviors	Elevated plus maze, Open field, Tail suspension	[35]
	miR-494	Amygdala (M Sprague-Dawley rats)	Via regulation of chromatin structure in the central nucleus increase anxiety-like behaviors	Elevated plus maze	[36]
**U**K**F**	miR-132 miR-212	Hippocampus, Amygdala (C57BL/6 M/F mice)	Stress-inducing paradigms alter their expression	Elevated plus maze, Open field	[37]

M: Male, F: Female, H: Human, UKF: Unknown Function.

## Data Availability

Data included in article/referenced in the report.

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
