# Peer review of "MAPK Is a Mutual Pathway Targeted by Anxiety-Related miRNAs, and E2F5 Is a Putative Target for Anxiolytic miRNAs"

_biomolecules, 2023, doi:10.3390/biom13030544_

Round 1

Reviewer 1 Report

The manuscript "The mitogen-activated protein kinase (MAPK) mutual signalling pathway targeted by anxiety-related miRNAs: cell cycle-related protein E2F5 is a putative target for anxiolytic miRNAS" by Amini et al., provides a very good and brief overview of the mechanisms of action of miRNAs that have been shown to have an anxiolytic or anxiogenic function. In addition, the authors present an association of the different ARmiRs with common target genes and discuss the impact of the signalling pathways involved in neuropsychological diseases.

Nevertheless, the following points should be addressed:

Major concerns:

1. Paragraph 3.5: the MAPK pathway is listed for both anxiogenic and anxiolytic miRNAs. It is now unclear how one and the same pathway can initiate opposite effects. This should already be addressed in this section. Otherwise in discussion.

2. How can it be explained that the signaling pathways involved shown in 3.5 do not appear in the analysis in 3.6? Perhaps it does not make sense to look at "only" the selected miRNAs that are presented as "higher effects". The results illustrate the fact that it makes less sense to pick out individual target genes/miRNAs/proteins when researching diseases, as these do not necessarily reflect the overall picture of the disease.

This aspect should be included and discussed in the appropriate paragraph.

Minor concerns:

1. The title is much too long and should be shortened.

2. Throughout the text, there are some places where individual words are formatted differently. In particular, some words are larger in font (e.g. line 24, 67, 107,135 etc). This should be corrected and standardised.

3. Lines 19 and 73: What are ARmiRs? Explanation follows only at the end of the introduction.

4. Figure 1: Numbers in the caption do not match the numbers in the figure? This is confusing for the reader and should be adjusted.

5. Line 191: "The t" at the beginning of the sentence should be deleted.

6. Line 199: What are Dragomir? There is no explanation!

7. Line 222: 3.1.5: What is in vitro overexpression in amygdala? Not quite clear what is meant by this.

8. Line 241: Heading should state “anxiogenic”

9. Line 267: HRs - explanation?

10. Paragraph 3.6: SLC17A6 is correct. The entire paragraph refers to SLC176A. Please check.

11. Figure 3b: How can I identify that SLC17A6, WASL, ATXN1 have a "higher rank"? The proteins are distributed differently in the figure and it seems that there are many more proteins that could have a strong influence. Please explain this statement.

12. Figure 3c: Labelling of the proteins very small. Difficult to read. Please adapt.

13. Line 406: "foundeight8" - Please check

14. Line 439: "funding" - please check

15. Section 3.8: The authors talk about reporting expression of E2F5 (line: 445: So, we reported that E2F5 expresses in all parts of the brain compared with the cerebral cortex). If I understand correctly, this are data from Proteinatlas. So I would expect this source being referenced here. And what does the statement "compared with the cerebral cortex" mean here? What differentiation is referred to in the following? Please rephrase this paragraph so that the statement is clear.

16. Line 470-471: Citations are formatted differently

17. Line 480-482: The sentence structure is not correct. Please check this.

Author Response

Reviewer 1

The manuscript "The mitogen-activated protein kinase (MAPK) mutual signaling pathway targeted by anxiety-related miRNAs: cell cycle-related protein E2F5 is a putative target for anxiolytic miRNAS" by Amini et al., provides a very good and brief overview of the mechanisms of action of miRNAs that have been shown to have an anxiolytic or anxiogenic function. In addition, the authors present an association of the different ARmiRs with common target genes and discuss the impact of the signaling pathways involved in neuropsychological diseases.

 Nevertheless, the following points should be addressed:

Major concerns:

  1. Paragraph 3.5: the MAPK pathway is listed for anxiogenic and anxiolytic miRNAs. It is now unclear how one and the same pathway can initiate opposite effects. This should already be addressed in this section. Otherwise in the discussion.

We appreciate this valuable suggestion. We have now included the paragraph describing this concern in the discussion part.

  1. How can it be explained that the signaling pathways involved shown in 3.5 do not appear in the analysis in 3.6? Perhaps it does not make sense to look at "only" the selected miRNAs that are presented as "higher effects". The results illustrate the fact that it makes less sense to pick out individual target genes/miRNAs/proteins when researching diseases, as these do not necessarily reflect the overall picture of the disease. This aspect should be included and discussed in the appropriate paragraph.

We appreciate this valuable suggestion. We have now included the paragraph describing this concern in the discussion part.

Minor concerns:

  1. The title is much too long and should be shortened.

We corrected it in the manuscript accordingly.

  1. Throughout the text, there are some places where individual words are formatted differently. In particular, some words are larger in the font (e.g. line 24, 67, 107,135 etc). This should be corrected and standardised.

We corrected it in the manuscript accordingly.

  1. Lines 19 and 73: What are ARmiRs? Explanation follows only at the end of the introduction.

Anxiety-related miRNAs (ARmiRs). We corrected it in the manuscript accordingly.

  1. Figure 1: Numbers in the caption do not match the numbers in the figure? This is confusing for the reader and should be adjusted.

We corrected it in the manuscript accordingly.

  1. Line 191: "The t" at the beginning of the sentence should be deleted.

We corrected it in the manuscript.

  1. Line 199: What are Dragomir? There is no explanation!

We corrected it in the manuscript.

  1. Line 222: 3.1.5: What is in vitro overexpression in amygdala? Not quite clear what is meant by this.

We corrected it in the manuscript.

  1. Line 241: Heading should state “anxiogenic”

We corrected it in the manuscript accordingly.

  1. Line 267: HRs - explanation?

We corrected it in the manuscript.

  1. Paragraph 3.6: SLC17A6 is correct. The entire paragraph refers to SLC176A. Please check.

We corrected it in the manuscript accordingly.

  1. Figure 3b: How can I identify that SLC17A6, WASL, ATXN1 have a "higher rank"? The proteins are distributed differently in the figure and it seems that there are many more proteins that could have a strong influence. Please explain this statement.

We corrected it in the manuscript accordingly (2.5.).

  1. Figure 3c: Labelling of the proteins very small. Difficult to read. Please adapt.

We corrected it in the manuscript accordingly.

  1. Line 406: "foundeight8" - Please check

We corrected it in the manuscript.

  1. Line 439: "funding" - please check

We corrected it in the manuscript.

  1. Section 3.8: The authors talk about reporting expression of E2F5 (line: 445: So, we reported that E2F5 expresses in all parts of the brain compared with the cerebral cortex). If I understand correctly, this are data from Proteinatlas. So I would expect this source being referenced here. And what does the statement "compared with the cerebral cortex" mean here? What differentiation is referred to in the following? Please rephrase this paragraph so that the statement is clear.

We corrected it in the manuscript accordingly.

  1. Line 470-471: Citations are formatted differently

We corrected it in the manuscript accordingly.

  1. Line 480-482: The sentence structure is not correct. Please check this.

We corrected it in the manuscript accordingly.

Reviewer 2 Report

Q: “ARDs are chronic and disabling diseases, the world's sixth leading cause of helplessness”.  What is the reference for this sentence? I check the WHO website, (https://www.who.int/news/item/30-03-2017--depression-let-s-talk-says-who-as-depression-tops-list-of-causes-of-ill-health) and a paper (https://www.ncbi.nlm.nih.gov/pmc/articles/PMC8157816/) shows that anxiety is the leading cause of ill health and disability worldwide. Please explain.

Q: line 48: normal to abnormal (subjects?)

Q:line 66: which means?

Q:line 73: ARmiRs? If this is the abbreviation of anxiety-related miRs, why in the line 75 you used anxiety-related genes? Please correct them to be consistent.

Q:line 90: Didn't" is more informal and more commonly used, whereas "Did not" is used in formal situations.

Q:line 91: were considered an exclusion criterion= were excluded.

Q:table1, you need a title, or a name for each column

Q:table1: at the bottom of the table you used UKF to represent Unknown Function, but in the table, you used UF

Q:line 164: full-name of let-7d  (The human lethal-7d)

Q:line 176: Feingold syndrome ? Please explain it in brief.

Q:  the format of the words in several lines (line 226, 253-261, 362-372, and many… etc,) is different. Please check.

Q: Figure 4:General description of the signaling pathways in highlighted anxiogenic miRNAs (miR- 400 19b, miR-92a, and 218). I suggest you to revise it as

Illustration the signaling pathways of anxiogenic (related?, in your previous line you used anxiety “related” gene ) miRNAs

Q:line 403: anxiolytic (related?) miRs

Q:line 406: foundeight8 ???

Q: line 403: Most influenced anxiolytic miRs in cellular pathways and their common targets

The expression style of this sentence is a little bit of “awkward”. It will be better if you can find an English native speaker to revise it.

Line 408: non??   Non”e” 

line 408 and 410, 411  interaction? Interaction”s”

line 415: represented? “re-drawed” may not confuse your readers in the future.

Q:figure 5:respect? What does this word mean here?

Q: Figure 6. The description of this figure please sees the previous suggestion.

Q: Figure 7:up=upper table    down= lower table

This paper needs an extensive English editing before being accepted.

Author Response

Reviewer 2

Q: “ARDs are chronic and disabling diseases, the world's sixth leading cause of helplessness”.  What is the reference for this sentence? I check the WHO website, (https://www.who.int/news/item/30-03-2017--depression-let-s-talk-says-who-as-depression-tops-list-of-causes-of-ill-health) and a paper (https://www.ncbi.nlm.nih.gov/pmc/articles/PMC8157816/) shows that anxiety is the leading cause of ill health and disability worldwide. Please explain.

We corrected it in the manuscript accordingly.

Q: line 48: normal to abnormal (subjects?)

We corrected it in the manuscript.

Q:line 66: which means?

We corrected it in the manuscript.

Q:line 73: ARmiRs? If this is the abbreviation of anxiety-related miRs, why in the line 75 you used anxiety-related genes? Please correct them to be consistent.

We corrected it in the manuscript accordingly.

Q:line 90: Didn't" is more informal and more commonly used, whereas "Did not" is used in formal situations.

We corrected it in the manuscript accordingly.

Q:line 91: were considered an exclusion criterion= were excluded.

We corrected it in the manuscript accordingly.

Q:table1, you need a title, or a name for each column

We corrected it in the manuscript accordingly.

Q:table1: at the bottom of the table you used UKF to represent Unknown Function, but in the table, you used UF

We corrected it in the manuscript accordingly.

Q:line 164: full-name of let-7d  (The human lethal-7d)

We corrected it in the manuscript accordingly.

Q:line 176: Feingold syndrome ? Please explain it in brief.

We corrected it in the manuscript accordingly.

Q:  the format of the words in several lines (line 226, 253-261, 362-372, and many… etc,) is different. Please check.

We corrected it in the manuscript accordingly

Q: Figure 4:General description of the signaling pathways in highlighted anxiogenic miRNAs (miR- 400 19b, miR-92a, and 218). I suggest you to revise it as Illustration the signaling pathways of anxiogenic (related?, in your previous line you used anxiety “related” gene ) miRNAs

We corrected it in the manuscript accordingly

Q:line 403: anxiolytic (related?) miRs

We corrected it in the manuscript accordingly

Q:line 406: foundeight8 ???

We corrected it in the manuscript accordingly

Q: line 403: Most influenced anxiolytic miRs in cellular pathways and their common targets The expression style of this sentence is a little bit of “awkward”. It will be better if you can find an English native speaker to revise it.

We corrected it in the manuscript accordingly

Line 408: non??   Non”e” 

We corrected it in the manuscript accordingly

line 408 and 410, 411  interaction? Interaction”s”

We corrected it in the manuscript accordingly

line 415: represented? “re-drawed” may not confuse your readers in the future.

We corrected it in the manuscript accordingly

Q:figure 5:respect? What does this word mean here?

We corrected it in the manuscript accordingly

Q: Figure 6. The description of this figure please sees the previous suggestion.

We corrected it in the manuscript accordingly

Q: Figure 7:up=upper table    down= lower table

We corrected it in the manuscript accordingly

This paper needs an extensive English editing before being accepted.

The manuscript was rechecked for typology and grammar errors and the quality of the text has been improved according to your comments.

Round 2

Reviewer 1 Report

The authors have implemented all the points addressed very well, so that the manuscript can be accepted for publication in its present form.

Author Response

Reviewer:

The authors have implemented all the points addressed very well so the manuscript can be accepted for publication in its present form.

Response:

Thank you very much for your time and valuable comments.

Reviewer 2 Report

Line 91, 95: didn't= did not

There are still some spelling errors or typos needed to be corrected.

Author Response

Reviewer:

Line 91, 95: didn't= did not

There are still some spelling errors or typos needed to be corrected.

Response:

We double-checked the manuscript and corrected mentioned errors and others thoroughly.